Functions of wheat ncRNAs in pathogen defense and stress adaptation

Yang Yongji
Hu Yi
Li Tao
You Qi youqi@yzu.edu.cn
Jiangsu Key Laboratory of Crop Genomics and Molecular Breeding/Zhongshan Biological Breeding Laboratory/Key Laboratory of Plant Functional Genomics of the Ministry of Education/Jiangsu Key Laboratory of Crop Genetics and Physiology, Agricultural College of Yangzhou University , Yangzhou , China
Nunes-da-Fonseca Rodrigo
Electronic publication date: 2025 Oct 22
Publication date: 2025
Volume: 13
Electronic Location ID: e20142
Received 2025 May 21; Accepted 2025 Sep 4
Copyright: ©2025 Yang et al.
Copyright year: 2025
Copyright holder: Yang et al.
License: This is an open access article distributed under the terms of the Creative Commons Attribution License, which permits unrestricted use, distribution, reproduction and adaptation in any medium and for any purpose provided that it is properly attributed. For attribution, the original author(s), title, publication source (PeerJ) and either DOI or URL of the article must be cited.
License URL: https://creativecommons.org/licenses/by/4.0/

Keywords: Wheat, Non-coding RNA, Disease, Stress, High-throughput data

Funding: Jiangsu Key R&D Program (Modern Agriculture) BE2023331 National Natural Science Youth Science Foundation of China 32000458 National Natural Science Foundation of China, International Cooperation 32261143462 This study was financially supported by the Jiangsu Key R&D Program (Modern Agriculture) Project No. BE2023331 and National Natural Science Youth Science Foundation of China, project: No. 32000458 and National Natural Science Foundation of China, International Cooperation, project: 32261143462. The funders had no role in study design, data collection and analysis, decision to publish, or preparation of the manuscript.

==============================
In organisms, non-coding RNAs (ncRNAs) are key regulatory elements that modulate, the expression of genes involved in diverse biological traits. Among them, micro RNAs (miRNAs), small interfering RNAs (siRNAs), and long non-coding RNAs (lncRNAs) have become major research focuses. Wheat, the world’s most widely grown crop, occupies 17% of global cultivated land and supplies ∼55% of the world’s carbohydrates. Understanding the roles, identification, and mechanisms of wheat ncRNAs is essential for both basic research and crop improvement. Through systematic searches of PubMed, Web of Science, and EndNote databases, this study identified 182 publications related to wheat ncRNAs. Based on predefined criteria—research relevance and publication timeframe (2015–2025)—70 high-quality studies were selected for in-depth analysis. This review comprehensively summarizes recent advances in ncRNA research (focusing on lncRNAs and small RNAs) in relation to wheat diseases, pests, and responses to biotic and abiotic stress. By integrating traditional classification with functional characterization, we developed a comprehensive analytical framework encompassing “molecular characteristics-biotic stress–abiotic stress”. Furthermore, this review consolidates multi-omics high-throughput data and online ncRNA databases. The integration of multi-omics technologies aims to provide both a theoretical foundation and novel strategies for wheat genetic improvement.

Types of Non-Coding RNA

A large portion of the eukaryotic genome consists of non-coding regions, and the RNAs transcribed from these regions are termed non-coding RNAs (ncRNAs), which are transcribed but not translated (Yu et al., 2019). ncRNAs are classified by length into long non-coding RNAs (lncRNAs) and small non-coding RNAs, the latter including microRNAs (miRNAs), small interfering RNAs (siRNAs), and others (Zhang et al., 2023b).

miRNAs single-stranded regulatory ncRNAs, typically 20–24 nucleotides (nt) long (D’Ario, Griffiths-Jones & Kim, 2017), with most plant miRNAs being 21 nt long (Zhan & Meyers, 2023). They were first discovered during a screen for abnormal development in Caenorhabditis elegans (Zhao et al., 2021b; Waterhouse & Hellens, 2015). Together with siRNAs, they are the most widely studied small RNAs in plant RNAs. In multicellular organisms, miRNAs regulate the expression of numerous protein-coding genes involved in growth and development throughout the life cycle (Rabuma, Gupta & Chhokar, 2022).

siRNAs were discovered earlier than miRNAs in plants. They mediate degradation or translational repression of target genes (Wu et al., 2020) and include several types such as trans-acting siRNAs (ta-siRNAs), natural antisense transcript siRNAs (nat-siRNAs), repeat-associated siRNAs (ra-siRNAs), and long siRNAs (lsiRNAs). Their mature sequence lengths vary. siRNAs produced by Dicer-like protein 1 (DCL1) and Dicer-like protein 4 (DCL4), including ta-siRNAs and nat-siRNAs (as well as miRNAs), are typically 21 nucleotides. Dicer-like protein 3 (DCL3) cleaves dsRNAs to produce 24-nt siRNAs, such as ra-siRNAs. Dicer-like protein 2 (DCL2) usually generates 22-nt siRNAs (Zhang et al., 2023b) and some nat-siRNAs also fall into this category. A few siRNAs range from 30–40 nt in length (Rabuma, Gupta & Chhokar, 2022).

Despite their relatively recent discovery, significant progress has been made in the study of plant lncRNAs, driven by advances in high-throughput technologies and bioinformatics. LncRNAs are a highly heterogeneous group of biomolecules, typically ≥200 nucleotides long, with little or no coding potential (Madhawan et al., 2020). Although most lncRNAs are transcribed by RNA polymerase II, some are transcribed by RNA polymerase III, and a few plant-specific lncRNAs are transcribed by RNA polymerases IV and V. Predominantly nuclear, lncRNAs often possess 5′ and 3′ structures similar to mRNAs. Compared to mRNAs, lncRNAs are generally shorter, contain fewer exons and introns, lack clear open reading frames, are expressed at lower levels, and show spatio-temporal and tissue-specific expression patterns. These features likely contribute to their non-coding nature and functional roles in organisms. Currently, there is no standardized classification system for lncRNAs. They are primarily categorized into five types—sense, antisense, bidirectional, intronic, and intergenic—based on their genomic location. Additional classifications are based on genomic features (e.g., promoter-, enhancer-, or transposon-associated) and mode of action, including interaction with proteins, DNA, or other RNAs (Huang et al., 2023; Yang et al., 2023).

NcRNAs in Response to Pests and Diseases in Wheat Crops

lncRNAs play key roles in plant defense against pests. Cagirici, Biyiklioglu & Budak (2017) conducted target analysis using wheat miRNA expression data in varieties resistant to Wheat Stem Sawfly (WSS), available in the Plant Database Collection. They identified larval-specific miR-87 as targeting a locus on chromosome 5BL in larvae, and miR-281 as targeting a transcript on chromosome 2AL in both male adults and larvae. These findings suggest that larval miRNA targets may contribute to pest resistance in wheat.

ncRNAs also play vital roles in plant-pathogen interactions. lncRNAs participate in various biological processes, and RNA interference (RNAi)-mediated silencing of pathogenic genes has been shown to inhibit filamentous fungal fungus, offering a promising strategy for disease control. Table 1 summarizes ncRNAs and their regulatory mechanisms in wheat under different disease conditions.

Table 1 ncRNAs in wheat diseases and stress responses.

Function	ncRNAs	Reference(s)	
Regulation of resistance
to Fusarium Head Blight	lncRsp1
miR398
miR1432
XLOC_302848
XLOC_321638
XLOC_113815
XLOC_123624	Wang et al. (2021), Wang et al. (2022), Wu et al. (2025)	
Regulation of rust
resistance	TCONS_00155902
TCONS_00103472
TCONS_00147277
TCONS_00147276
TCONS_00029083
TCONS_00013986
novel-Ta-miR02	Chen et al. (2015), Das et al. (2023)	
Regulation of powdery mildew resistance	Ta-miR397
lncRNA MSTRG.20701	Nair et al. (2022), Guan et al. (2023)	
Modulation of blight resistance	lncRNA MSTRG.4380.1	Cao et al. (2023)	
Pest defense	miR-87
miR-281	Yang et al. (2023)	
Modulating co-infection by both viruses (TriMV and WSMV)	miR397-5p
miR398
miR9670-3p	Yi et al. (2023)	
Regulation of drought stress in wheat	miR159
Ta-miR408
miR166 h
miR9662a-3p
Ta-miR5062-5A
novel-miR-340
novel-miR-417
miRN4330
miR5071a
miRN4321b
#PS_199	Chen & Yu (2023), Akdogan et al. (2016), Zhou et al. (2024), Gómez-Martín et al. (2023), Sharma et al. (2025)	
Regulation of cold stress in wheat	Traes_2BS_7A04BF5D5
miR394	Hou et al. (2025), Diaz et al. (2019)	
Regulation of heat stress in wheat	miR1117
miR1125
miR1130
miR113
miR156
miR166
miR393
lncRNA MSTRG.20144
lncRNA MSTRG.31273
lncRNA MSTRG.51285	Song et al. (2017), Babaei, Bhalla & Singh (2024), Kumar et al. (2015)	
Regulation of salt stress in wheat	miR159a
miR160
miR167
miR1118	Song et al. (2017), Sharma et al. (2021), Qiao et al. (2023)	
Involved in cadmium transport and detoxification, photosynthesis and antioxidant defense	lncRNA37228
Ta-miR9670
Ta-lncRNA18313	Zhang et al. (2023a), Zhu et al. (2023), Ma et al. (2024	
Notes.

ncRNAs in disease/stress responses and reference(s).

The incidence of downy mildew reduces both yield and grain quality in crops such as wheat, barley, and maize, with serious implications for human and animal health. Recently, Baldwin et al. (2018) demonstrated that DON production and small RNA (sRNA) populations respond to RNAi silencing of TRI6, a transcription factor that positively regulates DON synthesis by controlling TRI5 expression. This was achieved through RNAi-based host-induced gene silencing (HIGS) in Aspergillus oryzae, a wheat germplasm bacterium engineered to express RNAi vectors encoding dsRNAs (Baldwin et al., 2018). LncRNAs play important roles in many biological processes, but their functions and mechanisms in filamentous fungi remain poorly understood. A novel antisense lncRNA, GzmetE-as, was identified in Fusarium graminearum and shown to regulate its target gene GzmetE, which controls asexual and sexual reproduction in Aspergillus erythropolis via the RNAi pathway, thereby influencing its pathogenicity (Wang et al., 2021).

Figure 1 Mechanism of non-coding RNA regulation in fusarium head blight of wheat.

lncRsp1, located 99 bp upstream of the sugar transporter gene Fgsp1 in Fusarium graminearum, works in concert with Fgsp1 to negatively regulate the deoxynivalenol (DON) biosynthesis genes TRI4, TRI5, TRI6, and TRI13, thereby reducing DON production and fungal virulence (Wang et al., 2022). During the early stage of F. graminearum infection (within 12 h), 96.6% of differentially expressed lncRNAs were specifically activated. At 24 h post-inoculation (hpi) in wheat, only a few remained active, while protease-related genes showed specific enrichment. Among these, XLOC_302848 and XLOC_321638 are wheat lncRNAs linked to erythrina resistance. Joint analysis of RNA-seq data and fine QTL mapping for wheat erythrina resistance identified XLOC_113815 and XLOC_123624 as candidate lncRNAs involved in wheat’serythrina resistance response (Duan et al., 2020). The regulation of gene expression plays a crucial role in plant defense against pathogenic bacteria. Figure 1 illustrates the regulatory mechanisms of Fusarium head blight (FHB) resistance in wheat.

Most up-regulated genes after pathogen infestation were annotated as disease resistance genes, suggesting that plants actively enhance the expression of these genes in response to infection. However, this response is not a simple gene switch but involves complex regulatory networks. For example, miRNAs play a key role in wheat’s response to Fusarium graminearum. Upon infection, most miRNAs show negative correlations with their target genes, implying that miRNAs may suppress target expression to modulate disease resistance. The core difference between WY and CS lies in their miRNA-mediated regulatory networks. WY exhibits strong resistance to FHB through precise regulation—such as downregulation of tae-miR1122a—which activates key defense pathways. In contrast, CS shows upregulation of novel miR228, which may suppress defense responses and shift its overall response toward basic metabolism, resulting in weaker disease resistance (Wu et al., 2025). miR398 showed significant expression upon FHB infection, influencing resistance by regulating reactive oxygen species (ROS) balance. Its dynamic, tissue-specific expression in susceptible cultivars (with temporal differences between roots and leaves) suggest that spatio-temporal ROS regulation plays a crucial role in FHB defense. Similarly, the wheat miR1432 family contributes to disease resistance through dual targeting: EF-hand calcium-binding proteins and 4D hexose transporter genes. Its high expression in both yellow rust and FHB infections highlights its conserved regulatory role across pathogen interactions (Muslu et al., 2025). Besides miRNAs, siRNAs are also involved in wheat’s resistance to wheat blast. Rhizoctonia solani suppresses siRNA pathways, leading to the upregulation of resistance genes. Downregulation of siRNAs may relieve RdDM (RNA-directed DNA methylation)-mediated repression of resistance genes, enhancing resistance to R. solani. For example, sir4748, sir2582, and sir423 show significant downregulation of TaDCL3 (Dicer-like 3) expression after infection with B. equisetum, likely due to suppression of TaDCL3, which in turn improves resistance (Jin et al., 2020; Chen et al., 2015). These findings offer new insights into plant disease resistance mechanisms and suggest novel approaches for disease control.

Stripe rust is a global fungal disease that poses a serious threat to wheat by destroying most of the photosynthetic tissues in leaves, leading to reduced yields and poor seed quality. By identifying susceptible and resistant lncRNAs and their expression in wheat infected with stripe rust. Through bioinformatics analysis, Das et al. (2023) found that lncRNAs can also serve as targets of wheat miRNAs that regulate multiple genes involved in key defense processes against stripe rust. Six lncRNAs—TCONS_00155902, TCONS_00103472, TCONS_00147277, TCONS_00147276, TCONS_00029083, and TCONS_00013986—act as endogenous targets of Ta-miR1127a, which regulates proteins encoding disease resistance or NBS-LRR proteins, among others, and are involved in stripe rust resistance, revealing a novel mechanism of lncRNAs as miRNA ‘sponges’ in wheat stripe rust resistance (Das et al., 2023). Novel-Ta-miR02 may also be a key regulator of the defense response to common wheat stem and leaf rust, and the differential expression patterns of miRNAs reveal stem- and leaf-rust-responsive miRNAs and their potential roles in balancing resistance and susceptibility (Nair et al., 2022). In conclusion, research into wheat–rust interactions continues to highlight the crucial role of ncRNAs in wheat disease resistance mechanisms.

Under powdery mildew infection, wheat Ta-miR397 negatively regulates resistance by targeting the wound-induced protein (Ta-WIP). Overexpression of Ta-miR397 significantly increases susceptibility to powdery mildew, accelerates fungal spore germination and hyphal growth, and promotes tillering. Silencing the target gene Ta-WIP also enhances disease susceptibility (Guan et al., 2023). Predicting and functionally annotating lncRNA target genes can identify lncRNAs that promote or inhibit wheat powdery mildew development. The co-expression patterns of lncRNAs with neighboring mRNAs suggest significant correlations with the expression patterns of their potential targets. TraesCS5D03G0595900 and TraesCS5B03G1159600 are involved in wheat–pathogen interactions, and sugar signaling contributes to wheat’s immune response by potentially acting as a signal to trigger defense mechanisms. Among them, lncRNA MSTRG.20701 regulates fructose and mannose metabolism and participates in plant defense, providing a foundation for further understanding the pathogenesis of wheat powdery mildew (Cao et al., 2023).

The investigation of wheat blight revealed that Rickettsia species lncRNAs and miRNAs are involved in the infection process, and lncRNA MSTRG.4380.1 can reduce the virulence of wheat blight pathogens, offering a new strategy for disease control (Yi et al., 2023). These studies underscore the important roles of noncoding RNAs in wheat’s response to various stresses, laying the groundwork for future research into their molecular mechanisms and applications.

With the advancements in research on wheat under dual-virus infection (TriMV and WSMV), 28 differentially expressed miRNAs (e.g., miR168a, miR397-5p, etc.) were identified as virus-responsive. Functional validation confirmed the regulatory roles of miR397-5p, miR398, and miR9670-3p during viral infection, with miR9670-3p acting as a negative regulator for both TriMV and WSMV infections. Target prediction showed these miRNAs primarily regulate genes involved in defense responses, catalytic activity, and nucleic acid binding (Soylu et al., 2024).

Figure 2 Non-coding RNA regulation of wheat growth, development, and stress conditions.

NcRNAs Associated with Growth and Development and Abiotic Stress Response in Wheat Crops

NcRNAs play regulatory roles in wheat growth and development, influencing agronomic traits as illustrated in Fig. 2. Li et al. (2019) conducted transcriptome analysis across four developmental stages (three-leaf, winter dormancy, spring green-up, and jointing) to identify and characterize miRNAs in wheat. They found that miR168 was highly expressed at all stages and identified eight miRNA target genes, two of which were conserved targets of miR171 and miR172, respectively. miR1172a targeted the disease resistance protein RGA1 (heterotrimeric GTP-binding protein α subunit), and miR9674b-5p targeted the Rf1 gene, among others, showing stage-specific differential expression. In addition, ncRNAs regulate the cell cycle and anther development. For instance, lncRNA_047461, lncRNA_074658, and lncRNA_061738 may be involved in cell cycle regulation (Ma et al., 2018). Ta-miR2275-3p is implicated in meiotic processes and early anther development in wheat (Sun et al., 2018). The lncRNA MSTRG.59353 targets AGO1d-7A and AGO1d-7B, contributing to spike shape and development (Cao et al., 2021). The interaction between miR172 and the Q allele reduces single nucleotide polymorphisms (SNPs) at the miRNA binding site. Inhibiting miR172 through target mimics results in a compact spike and transformation of glumes into florets in the apical spikelet (Debernardi et al., 2017). In studies of ncRNA-mediated regulation of wheat spike development, the cultivar Guomai 301 (wild type, WT) and its three spike mutants (drs, ass, and ptsd1) were compared. Ta-miR396b expression was significantly higher in WT than in drs and ass mutants. Dysregulation of TaGRFs by Ta-miR396b led to distinct phenotypes: drs had a dwarfed, rounded spike; ass showed impaired apical spikelet development causing sterility; and ptsd1 likely disrupted spike differentiation (Yao et al., 2024). Additional studies showed that Ta-miR397-6A and Ta-miR397-6B encode functional Ta-miR397a, whose expression increases during grain filling. Suppression of Ta-miR397a reduced grain size and weight, while its overexpression promoted grain filling and enhanced grain size and weight, indicating its role as a positive regulator of grain development (Wang et al., 2024).

Plant small RNAs are also involved in epigenetic processes and serve as key components of gene regulatory networks controlling development and homeostasis. They respond to abiotic stresses such as heat, cold, salinity, and dehydration. sRNA expression changed substantially depending on stress treatments. For example, miRNAs help fine-tune wheat growth and development. miR166a-b and miR167a-b show low nucleotide polymorphism across wheat species, suggesting conserved and vital regulatory roles in wheat growth and development (Singh et al., 2020). Chen & Yu (2023) identified three endoplasmic reticulum (ER) stress-responsive miRNAs (Ta-miR164, Ta-miR2916, and Ta-miR3.6e−5p) using integrated miRNA sequencing and degradome analysis. These miRNAs and their targets responded to stressors like DTT, PEG, NaCl, and temperature extremes. Using a BSMV-based silencing system, suppression of these miRNAs significantly improved wheat tolerance to drought, salt, and heat.

ncRNAs participate in diverse biological processes across species, including plant growth, development, and biotic and abiotic stress responses. To cope with environmental challenges, such as drought, heat, cold, and salinity, plants develop complex regulatory mechanisms involving ncRNAs. Known and newly identified ncRNAs are summarized in Table 1 according to their functions.

Drought is the most common environmental stress that inhibits crop growth and development, severely limiting global wheat production. ncRNAs can influence plant drought tolerance by regulating the expression of target genes (Li et al., 2022). Drought-responsive miRNAs have been reported in Arabidopsis thaliana, Oryza sativa, and Glycine max (Wang et al., 2016), but relatively few have been identified in wheat. Research shows that the drought-tolerant wheat cultivar Sivas 111/33 exhibits distinct miRNA regulatory patterns under drought stress: several miRNAs (e.g., miR156, miR159, and miR398) are upregulated to enhance stress response, while others (e.g., miR164 and miR482) are downregulated, accompanied by upregulation of their target genes, suggesting involvement in stress resistance via inverse regulation. In contrast, the drought-sensitive cultivar Atay 85 displays weaker miRNA expression changes and abnormal regulation of target genes (e.g., downregulation of miR5048 targets), potentially contributing to its lower drought tolerance. Notably, under drought stress in leaves, the significant upregulation of miR159 targets ABA-positive regulators MYB33 and MYB101 transcription factors, enhancing drought tolerance by modulating ABA accumulation. These findings highlight key differences in miRNA regulatory networks between wheat cultivars with varying drought resistance (Akdogan et al., 2016). Li et al. (2022) identified roles for differentially expressed lncRNAs, miRNAs, and their target genes in wheat drought tolerance, discovering a drought stress-responsive lncRNA–miRNA–mRNA regulatory module. They hypothesized that regulatory modules centered on novel-miR-340 and novel-miR-417 regulate different drought-tolerance genes, thereby conferring distinct drought resistance traits to wheat. During the grain-filling stage under drought stress, Ta-miR408 enhances photosynthetic efficiency and antioxidant capacity by targeting and suppressing allene oxide synthase (AOS) genes, thereby inhibiting jasmonic acid (JA) and abscisic acid (ABA) biosynthesis (Zhou et al., 2024). Deep sequencing revealed downregulation of miR166 h and upregulation of miRN4330 in roots, along with downregulation of miRN4321b and upregulation of miR5071 in leaves under drought stress. These miRNAs mediate stress adaptation by targeting key genes, including HD-ZIP III transcription factors (developmental regulators) and Mla1 (a disease resistance gene). Notably, miRN4330 may regulate endoplasmic reticulum metabolism, miR5071a targets NB-LRR domain-containing disease resistance proteins, and miRN4321brepresents a novel function reported for the first time in wheat (Gómez-Martín et al., 2023). Other studies identified 306 known and 58 novel miRNAs in two wheat genotypes, with miR9662a-3p showing the highest expression. Quantitative reverse transcription polymerase chain reaction (qRT-PCR) confirmed differential expression of 10 novel miRNAs under drought stress; among them, #PS_199 showed significantly higher expression in the roots of the NI5439 genotype, suggesting a role in root-specific drought response mechanisms (Sharma et al., 2025). Under drought and salt stress, Ta-miR5062-5A expression was downregulated, while its target gene TaCML31 (a calmodulin-like gene) was upregulated. Studies revealed that TaCML31 interacts with the MYB transcription factor TaMYB77, and together they regulate osmotic protectant accumulation, stomatal closure, root development, and ROS homeostasis. TaMYB77 activates expression of stress defense genes TaP5CS2, TaNCED1, and TaDREB3 by binding to their promoters. These findings indicate that the Ta-miR5062-5A-Hap1 haplotype enhances wheat drought resistance (Hou et al., 2025).

Cold stress alters the expression of lncRNAs and miRNAs in wheat. LncRNAs can function as miRNA targets. For example, lncRNA Traes_2BS_7A04BF5D5 responds to cold stress (Diaz et al., 2019). The target genes of differentially expressed miRNAs are mainly associated with stimulus response, transcriptional regulation, and ion transport. For instance, under cold stress, differential expression of miR5169 may affect iron transport proteins. In wheat, 39 miRNAs from 28 families show differential expression under cold treatment, including miR394, which also responds to cold stress (Song et al., 2017).

lncRNAs regulate wheat pollen development under heat stress by acting in cis, trans, or as miRNA repressors. Babaei, Bhalla & Singh (2024) found interactions between lncRNAs and miRNAs under heat stress. Heat-responsive miRNAs—such as miR1117, miR1125, miR1130, and miR113—along with lncRNAs MSTRG.20144, MSTRG.31273, and MSTRG.51285, appear to influence pollen development via a ceRNA mechanism (Kumar et al., 2015). Ravichandran et al. (2019) showed that conserved members of the miR156, miR166, and miR393 families target SPL (SQUAMOSA PROMOTER BINDING PROTEIN-LIKE) transcription factors, HD-Zip-like transcription factors, and TIR1 (TRANSPORT INHIBITOR RESPONSE 1)/AFB (AUXIN SIGNALING F-BOX). Responsive gene-based SSR (cg-SSR) and miRNA gene-based SSR (miRNA-SSR) molecular markers (Sharma et al., 2021) may aid in marker-assisted breeding for heat-tolerant wheat varieties and the genetic diversity analysis of germplasm.

Salt stress significantly affects wheat growth. Eight miRNAs, including miR159a, miR160, and miR167, are significantly upregulated under salt stress and are suggested to mediate salt signaling through post-transcriptional or translational regulation, thereby modulating hormonal signaling pathways (Babaei, Bhalla & Singh, 2024). miR1118 exhibits salt stress sensitivity in wheat roots, with its expression significantly reduced under salt stress conditions, suggesting a role in seedling salt stress adaptation (Qiao et al., 2023). It post-transcriptionally cleaves and represses TaCaM2, which interacts with the transcription factor TaMYB44 to form a functional complex. TaMYB44 directly activates stress-responsive genes (TaGS2.2, TaNRT2.1, and TaPIN4), enhancing wheat tolerance to low nitrogen (Zhang et al., 2025).

Cadmium, a toxic and widespread metallic pollutant, disrupts plant growth, reducing yield and quality, and poses a threat to food safety and human health (Zhang et al., 2023a). Zhu et al. (2023) integrated physiological, biochemical, and RNA sequencing analyses to reveal a cadmium response mechanism mediated by lncRNAs in wheat. These lncRNA targets act via cis-regulation, influencing neighboring genes involved in cadmium transport, detoxification, photosynthesis, and antioxidant defense. lncRNA37228 and its target gene TraesCS4B02G159100 play key roles in cadmium resistance. The wheat-specific Ta-miR9670 enhances cadmium tolerance by targeting mitochondrial transcription termination factor (mTERF) genes. Overexpression of Ta-miR9670significantly increases seedling biomass while reducing malondialdehyde, hydrogen peroxide, and cadmium levels (Ma et al., 2024). Ta-lncRNA18313 is broadly expressed in leaves and strongly induced by cadmium stress. RNA-seq analysis identified 370 differentially expressed genes enriched in transcriptional regulation and antioxidant defense pathways. These findings suggest that Ta-lncRNA18313 improves cadmium tolerance by modulating oxidative stress and related gene expression (Zhao et al., 2025).

The Development of Multi-Omics Technologies Has Enhanced the Study of ncRNAs in Wheat

With the rapid advancement of science and technology, biological research has shifted from classical morphology and physiology to molecular biology. In this process, multi-omics approaches—including genomics, transcriptomics, metabolomics, and proteomics—have provided diverse data sources to support studies on wheat ncRNAs and offer new perspectives on the internal mechanisms of organisms.

A search of wheat data in the NCBI database reveals 44 Genome RefSeq assemblies, including reference genomes for varieties such as Chinese Spring, Kariega, Fielder, and Attraktion. There are 1,260 DNA and RNA BioProjects and 57,812 BioSamples, including 506 DNA and 754 RNA BioProjects. Among 1,219 second-generation sequencing datasets, most were generated using Illumina, while 35 were obtained using third-generation technologies such as PacBio. Collectively, these data form a strong foundation for research on wheat non-coding regions. Table 2 presents the number of entries in each category of high-throughput sequencing data from the NCBI public database.

Table 2 Classification and quantity statistics of wheat data in NCBI public databases.

Data classification	Bioproject	BioSample	
RNAseq	710	15,327	
snRNAseq	1	1	
ncRNAseq	6	284	
lncRNAseq	3	25	
miRNAseq	42	656	
mRMAseq	1	24	
sRNAseq	1	36	
Genome	136	69,357	
Epigenetic group	64	4,972	
Metabonomics	1	148	
Ribo-seq	3	18	
The Genome of related to stress	87	8,943	
Notes.

The first column lists the types of high-throughput sequencing data; the second column shows the number of BioProjects for each type; the third column shows the number of BioSamples.

The number of platforms and databases related to ncRNA is growing. The EVLncRNAs database (https://www.sdklab-biophysics-dzu.net/EVLncRNAs1/) integrates information from four small lncRNA databases (lncRNAdb, LncRAN-Disease, Lnc2Cancer, and PLNIncRBase), and contains 1,543 lncRNAs from 77 species, including 428 from 44 plant species (Zhou et al., 2019). Other plant ncRNA databases develpoed in the past three years include PLncDB V2.0 (Jin et al., 2021), NONCODE6 (Zhao et al., 2021a), CANTATAdb 3.0 (Szczesniak & Wanowska, 2024), PlantIntronDB (Wang et al., 2023), JustRNA (Tseng et al., 2023). ncPlantDB (https://bis.zju.edu.cn/ncPlantDB/), launched in 2025, is a newly established plant ncRNA database (Liu et al., 2025). The PlantCircRNA database integrates data from AtCircDB and CropCircDB, including tissue-specific and stress-responsive data, and features a novel nomenclature system (He et al., 2025). It is expected to offer valuable insights into plant circRNA research. The Rfam database (https://rfam.org/), which provides ncRNA families for genome annotation, has been updated to version 15.0 and now includes 26,106 genomes, including viral genomes, to enhance annotation quality (Ontiveros-Palacios et al., 2025). In 2023, Zhang et al. developed Triticeae-BGC, a web-based platform for the detection, annotation, and evolutionary analysis of wheat biosynthetic gene clusters. The platform, which is freely available online (http://119.78.67.240:3838/Triticeae-BGC/), allows inter-gene covariance analysis using gene or chromosome positions to identify additional candidate genes for further study (Li et al., 2023). WheatOmics (http://wheatomics.sdau.edu.cn/) is a multi-omics platform designed to accelerate functional genomics research in wheat. It supports gene identification (forward and reverse genetics), gene expression analysis, molecular network construction, regulatory element analysis, and the identification of superior haplotypes. The integrated GeneHub tool allows users to retrieve multi-omics data for individual wheat genes with a single click (Ma et al., 2021). Currently, there is no dedicated database specifically for wheat ncRNAs. While WheatOmics includes some lncRNAs and miRNAs, its functional annotations of ncRNAs remain limited.

Researchers have developed several ncRNA analysis tools to mine data from high-throughput sequencing, including miRnovo (Vitsios et al., 2017), miRkwood (Guigon et al., 2019), and miRDeep-P2 (Kuang et al., 2019). Additionally, sRNAminer enables rapid and accurate annotation of small RNAs (Li et al., 2024). For lncRNA prediction, LncCat identifies lncRNAs based on features such as ORF length (Feng et al., 2023). A novel computational tool, PlantLncBoost, has also been developed, significantly improving lncRNA prediction accuracy and cross-species generalization (Tian et al., 2025).

Summarizing and Looking Forward

While studies on non-coding RNAs have advanced in animal research, they remain underexplored in plant sciences, and their regulatory mechanisms are still poorly understood.

Current research on wheat non-coding RNAs in disease resistance faces three major challenges. First, functional validation lags behind, with most ncRNAs limited to bioinformatic predictions and lacking genetic validation through methods like CRISPR knockouts or overexpression. Second, existing data are constrained by cultivar and pathogen strain specificity, often based on single strains or wheat varieties, highlighting the need for broader analyses across different races and resistance backgrounds. Third, translational application faces bottlenecks, with limited progress in converting ncRNA findings into practical field-based disease-resistant breeding strategies. To overcome these challenges, future efforts should focus on: establishing a wheat ncRNA functional database incorporating data on Fusarium head blight, stripe rust, and powdery mildew; developing novel ncRNA delivery technologies; and exploring the co-evolutionary relationships between ncRNAs and QTLs. These directions will facilitate the practical application of ncRNAs in wheat disease-resistance breeding.

Current research on wheat non-coding RNAs in abiotic stress responses also faces several critical scientific questions that need urgent resolution. First, spatiotemporal expression patterns of ncRNAs remain unclear; for example, while drought-induced expression differences have been observed, underlying regulatory mechanisms such as tissue-specific splicing are not well understood. Second, the crosstalk between ncRNAs and plant hormone signaling networks is insufficiently studied, with many interaction mechanisms yet to be clarified. Third, translational challenges persist, as most findings remain at the laboratory stage without field validation. To address these gaps, future work should focus on elucidating ncRNA–hormone interactions, constructing spatiotemporal expression networks, and building a comprehensive wheat ncRNA functional database integrating multiple abiotic stress datasets. These advances will help bridge the gap between basic ncRNA research and its agricultural application.

Bioinformatics faces significant challenges in ncRNA research, particularly the limited accuracy of functional prediction. Current tools show poor adaptability to polyploid crops like wheat and lack specific algorithms for different ncRNA types, limiting the precision of functional annotation. Moreover, integrative multi-omics analyses remain underdeveloped, as most studies focus on single-omics levels and fail to systematically incorporate ncRNA data with transcriptomic, epigenomic, and proteomic datasets. This hampers a comprehensive understanding of ncRNA regulatory networks. To address these challenges, future research should pursue two key directions: first, improving database construction by developing a comprehensive wheat ncRNA functional database that integrates multi-stress and multi-cultivar data, along with species-specific prediction tools. Second, deepen mechanistic insights by using single-cell sequencing technologies to clarify the spatiotemporal expression patterns of ncRNAs and unravel their synergistic regulation with epigenetic modifications. These systematic efforts will significantly enhance the depth and practical value of wheat ncRNA research, offering new theoretical foundations and technical support for molecular design breeding.

Additional Information and Declarations

Competing Interests

Author Contributions

Data Availability

The authors declare there are no competing interests.

Yongji Yang performed the experiments, analyzed the data, prepared figures and/or tables, and approved the final draft.

Yi Hu performed the experiments, prepared figures and/or tables, and approved the final draft.

Tao Li conceived and designed the experiments, authored or reviewed drafts of the article, and approved the final draft.

Qi You conceived and designed the experiments, authored or reviewed drafts of the article, and approved the final draft.

The following information was supplied regarding data availability:

This is a literature review.

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
