# Peer review of "Functions of wheat ncRNAs in pathogen defense and stress adaptation"

_PeerJ, doi:10.7717/peerj.20142_

## Round 0.1 · original submission · Major Revisions

Based on the detailed evaluations from the four reviewers and the attached annotated manuscript (review-1-2084996.pdf), I recommend a decision of Major Revisions for the manuscript titled "Progress in the study of the functional mechanism of non-coding RNAs in wheat." Below is a summary of the key issues raised and detailes required for revision:

Decision: Major Revisions

Summary of Reviewer Comments

Reviewer 1 ( Annotated PDF)
Language & Grammar: Numerous issues with grammar, phrasing, and redundancy. Abstract: Repetitive, vague; lacks specifics (e.g., number of studies reviewed, timeframe). Terminology: Informal phrasing such as “simply turning gene expression on/off.” Figures & Tables: Not sufficiently described in the text.
Conclusion: Lacks discussion of research gaps and future directions.
References: Formatting inconsistencies; some citations lack full details.
Recommendation: Language editing + major structural and stylistic revisions.

Reviewer 2 (Rejection)
Literature Review Weaknesses: Does not sufficiently advance the field.
Citations: Many outdated or missing references, particularly recent wheat microRNA studies. Novelty: No substantial contribution beyond existing literature. Recommendation: Rewrite from scratch with updated references and clear added value.

Reviewer 3 (Major Revisions)
Formatting Issues: Inconsistent font sizes, word breaks, spacing.
Taxonomy: Species names not italicized. Technical Issues: Missing references for datasets, vague or redundant phrases.
Figures: Figure 1 not informative; conclusions section repetitive.
Recommendation: Rework structure, fix formatting, and improve scientific clarity.

Reviewer 4 (Major Revisions)
Title: Too general; should reflect specific focus on wheat ncRNAs and stress/disease context. Terminology Standardization: Correct inconsistent use of terms (e.g., “long-chain” vs. “long noncoding RNA”).
Scientific Clarity: Several misleading or inaccurate descriptions (e.g., miRNA vs. lncRNA roles). Conclusion: Lacks novelty, perspective, and discussion of gaps or future directions. Update Required: Must include new studies from 2025.
Recommendation: Substantial rewriting with deeper analysis and clearer structure.

Required Revisions Summary
Language & Grammar Submit to a professional English language editing service.
Abstract Include number of articles reviewed, date range, and clearer summary of content.
Title Make more specific to wheat ncRNAs and their roles in stress/disease.
Figures & Tables Reference and explain all visuals directly in the text.
Terminology Standardize (e.g., “lncRNAs” not “long-chain”; “miRNA” not “MiRNA”)
Conclusion Add clear future directions and identified knowledge gaps.
Citations Replace outdated references and add key recent studies (esp. from 2025).
Formatting Italicize species names, fix inconsistent fonts, remove line-break errors.
Scientific Clarity Clarify technical language (e.g., regulatory mechanisms, gene names).
Dataset References Properly reference public datasets and bioinformatics tools mentioned.

You are encouraged to thoroughly revise the manuscript by:
Rewriting key sections (Abstract, Results, Discussion, and Conclusion).
Conducting a comprehensive literature update with recent studies from 2024–2025.
Improving formatting, figures, and technical accuracy.
Enhancing the depth and critical synthesis of the review.
Once these changes are made, the manuscript may be suitable for reconsideration.

**Language Note:** The review process has identified that the English language must be improved. PeerJ can provide language editing services - please contact us at [email protected] for pricing (be sure to provide your manuscript number and title). Alternatively, you should make your own arrangements to improve the language quality and provide details in your response letter. – PeerJ Staff

Reviewer 1 ·

Basic reporting

see the attachment

Experimental design

-

Validity of the findings

-

Annotated reviews are not available for download in order to protect the identity of reviewers who chose to remain anonymous.

Reviewer 2 ·

Basic reporting

The progress and critical roles of ncRNA research in wheat are growing so fast. NGS helped and has expanded the ncRNA repertoire known in wheat, while functional studies have pinpointed specific ncRNAs influencing agronomic traits. Hence, this is unfortunately not adding much to the existing literature or community. There are many citations in wheat microRNA missing and too old one has to be replaced with the new one. This MS needs to be rewritten from scratch with the correct citations.

Experimental design

Needs to be improved with new/correct citations.

Validity of the findings

There are still areas to improve such as comparing with the existing repertoire in wheat.

·

Basic reporting

The manuscript provides an interesting resource for non-coding RNA in wheat.

At present the manuscript is not ready for progressing to publication:
* Problem with formatting manuscript re breaking up words in the right-side margin as well as missing a space after full stops (for example).
* Species names need to be in italics.
* Inconsistent font size suggests cut-paste from different sources.

Line 53 cite P Waterhouse, NATURE 520, 2 APRIL 2015, p42, gives a more balanced view of siRNA
Line 142 define TaDCL3, including the Traes ID
Line 175 needs a reference for the Traes information presented
Line 191 “whole-life” - this term is not clear in what is meant
Line 289 reference the “PRJNA1067657 project” is required. If this is not available, remove a section of MS to line 306
Line 340 should read “Research on non-codingRNA has made ….”
Lines 340 – 372. This section basically repeats the preceding text and is not particularly useful
Figure 1 is not useful
Figure 2 and Tables 1 and2 are a useful resource

Experimental design

The review seems to be an internal report that has been submitted and needs quite a bit of editing.

Validity of the findings

As noted, some information is not properly referenced.

Additional comments

The manuscript is not ready for publication.

·

Basic reporting

In the current review article authors performed a comprehensive literature study of non-coding RNAs in wheat. Wheat is one of the widely grown and major staple cereal crops with global significance. Given that a major part of the genome comprises non-coding RNAs, exploring the potential of non-coding RNA holds substantial promise for enhancing crop performance and developing disease-resistant wheat varieties. The review article outlines the non-coding RNA related to wheat disease as well as their significant role in biotic and abiotic stresses like heat, drought etc., Moreover, authors have reviewed the recent advances in multi-omics data and the relevant tools available to study the mechanistic role of these non-coding RNAs. Therefore, the concept and the theme of the review article hold importance and are highly relevant for the scientific audience. However, in the current version, it fails to capture audience interest and attention. More than just summarizing, the article lacks a fresh viewpoint. The manuscript needs to be thoroughly revised before being considered for publication. A thorough revision incorporating deeper analysis, sharper conclusions, and a clearer forward-looking perspective will greatly enhance its value and appeal to readers. To improve the review's impact and suitability for publication, I recommend the following revisions:

1. The Title is very general and is too broad. It doesn't fully reflect the article’s specific focus. Since the review highlights wheat’s non-coding RNAs in the context of disease and both biotic and abiotic stress tolerance, a more precise title would better guide the reader.

2. Line no. 48: The authors mentioned lncRNAs as long-chain non-coding RNAs. I think using the term chain is not suitable. The universally accepted term is “long noncoding RNAs (lncRNAs)” without “chain.” Similarly, the correct term is “small noncoding RNAs” not “small chain.”

3. Line 51: MiRNA should be written as miRNA throughout the whole manuscript.

4. Line 54: While using the term sRNA, does the author mean to say small RNAs? If yes, then the abbreviation should be written in full.

5. Line 61: Sometimes authors write siRNA and sometimes siRNA. The manuscript should standardize the usage of “siRNA” throughout. Uniformity should be maintained.

6. Line 140: What is RdDM? The pathway is misspelled. Authors must check for typos and misspellings very carefully throughout the article.

7. While writing the scientific name of the organism, the entire scientific name (both genus and species) should be italicized throughout the manuscript. For example, Rhizoctonia solaniis in line 141.

8. Authors must check carefully the space between two words so that there is no confusion in the readability. For example, the distance between “disease resistance” in line 167.

9. Line 248: Authors states “Heat-responsive lncRNAs, such as miR1117, miR1125, miR1130, and miR113”. I guess it should be Heat-responsive miRNAs…

10. Figure 2 is nowhere mentioned in the manuscript.

11. The conclusion is underdeveloped and does not offer clear future directions or highlight existing research gaps. The authors must discuss research gaps and unresolved issues.

Experimental design

This review article acknowledges the work before 2025 in wheat. There has been some significant research this year. Therefore, the authors should update the article to incorporate these 2025 findings and reflect the latest developments. For example, a report on ncRNA in wheat development and stress (Wang et al., 2025; Zhao et al., 2025; Muslu et al., 2025, etc.,)

Validity of the findings

-

Additional comments

Lastly, the whole article is poorly written. The English language requires refinement throughout the article. Sometimes it is very difficult to follow and understand.

---

## Round 0.2 · accepted · Accept

Congratulations for the acceptance of your manuscript.

Reviewer 1 ·

Basic reporting

No Comment

Experimental design

No Comment

Validity of the findings

No Comment

Additional comments

All comments and suggestions listed have been followed and meet publication standards.

·

Basic reporting

The authors have considerably worked on the language and readability of the manuscript. The manuscript is now clear, concise, and easy to follow, which enhances the overall quality of the work.

Experimental design

No Comments

Validity of the findings

The conclusion has been strengthened by incorporating the latest findings and outlining future research directions. These revisions have enhanced the overall clarity, relevance, and impact of the manuscript.

Additional comments

The authors have significantly improved the quality of the manuscript. This review comprehensively summarizes recent advances in ncRNA research (focusing on lncRNAs and small RNAs) in relation to wheat diseases, pests, and responses to both biotic and abiotic stresses. Overall, this review not only enhances current knowledge but also guides new research directions and strategies for wheat improvement. Therefore, I recommend this manuscript for publication.